

# Irritant and repellent behaviors of sterile male *Aedes aegypti* (L.) (Diptera: Culicidae) mosquitoes are crucial in the development of disease control strategies applying sterile insect technique

Wasana Boonyuan[1], Amonrat Panthawong[2], Thodsapon Thannarin[1], Titima Kongratarporn[1], Vararas Khamvarn[1], Theeraphap Chareonviriyaphap[2] and Jirod Nararak[2]

[1] Nuclear Technology Research and Development Center, Thailand Institute of Nuclear Technology (Public Organization), Nakhon Nayok, Thailand
[2] Department of Entomology, Faculty of Agriculture, Kasetsart University, Bangkok, Thailand

## ABSTRACT

The mosquito *Aedes aegypti*, known to transmit important arboviral diseases, including dengue, chikungunya, Zika and yellow fever. Given the importance of this disease vector, a number of control programs have been proposed involving the use of the sterile insect technique (SIT). However, the success of this technique hinges on having a good understanding of the biology and behavior of the male mosquito. Behavioral responses of *Ae. aegypti* male populations developed for SIT technology were tested under laboratory conditions against chemical and natural irritants and repellents using an excito-repellency (ER) chamber. The results showed that there were no significant behavioral escape responses in any of the radiation-sterilized male *Ae. aegypti* test populations when exposed to citronella, DEET, transfluthrin, and deltamethrin, suggesting that SIT did not suppress the expected irritancy and repellency (avoidance) behaviors. The type of information reported in the current study is vital in defining the effects of SIT on vector behavior and understanding how such behavior may influence the success of SIT technology with regard to other vector control interventions.

## INTRODUCTION

*Aedes* (Stegomyia) *aegypti* (L.) mosquito transmits key arboviral infections, including dengue, chikungunya, yellow fever and Zika viruses, which are serious public health concerns in various parts of the world, including Thailand (*Lwande et al., 2020*; *Raksakoon & Potiwat, 2021*). *Aedes aegypti* exhibits anthropophilic behavior (prefers feeding on humans) and a tendency to feed multiple times during an egg-laying cycle—which imparts this particular vector with a remarkable efficiency in pathogen transmission (*Scott & Takken, 2012*). This mosquito is endophilic (*i.e.,* it seeks shelter indoors) and endophagic

Corresponding author
Jirod Nararak, fagrjrn@ku.ac.th, jirod.nar@ku.th

(*i.e.,* it feeds on blood indoors), and exhibits movement patterns between indoor and outdoor environments.

Changing environmental factors or applying chemicals that target flying adults can reduce mosquito populations (*Peter et al., 2005*). However, it seems that based on the successes and failures of mosquito control, many insecticides seem to have reached the end of their effectiveness as a stand-alone strategy (*Rezende-Teixeira et al., 2022*). Thus, insecticides play the most important role in the global control of disease vector mosquitoes. Unfortunately, several species of mosquitoes have developed tolerance or resistance to many chemical insecticides, especially *Ae. aegypti* which has been found resistant to pyrethroids across Thailand (*Chareonviriyaphap et al., 2013*; *Liu, 2015*). The use of various approaches simultaneously is the current recommended vector control strategy and is a cornerstone of integrated vector management—a best-practice framework for long-term and cost-effective vector control (*Golding et al., 2015*).

The sterile insect technique (SIT) is an autocidal control approach that is commonly used to suppress or eradicate populations of some of the most important insect pests in agriculture, livestock, and human health (*Guo et al., 2022*). The SIT is based on the mass production of the target species, ionizing radiation sterilization of males, and the sustained and methodical release of large numbers of sterile individuals into the target area (*Gouagna et al., 2020*). For mosquito control, the SIT relies on rearing, sterilizing, and releasing large numbers of male mosquitoes that will mate with fertile wild females, thus reducing the production of offspring from the target population. The primary focus of research assessing the efficacy of release strategies, particularly those employing SIT technology, primarily revolves around impact evaluation on mating competitiveness relative to unaltered populations. Additionally, the overall fitness of SIT males, encompassing their capacity to survive and reproduce in the natural environment, is also examined within field conditions. (*Oliva et al., 2021*). *Amos et al. (2022)* demonstrated that both male and female *Ae. aegypti* shows attraction to humans at short range. Hence, it is possible that male mosquitoes may have an increased likelihood of encountering insecticides and repellents. The utilization of insecticides or repellents has the potential to impact the behavioral patterns of male mosquitoes.

Examining the influence of irritants and repellents on male behavior during home ingress and egress constitutes a pivotal dimension that necessitates prioritization for the optimization of this vector management methodology (*Takken & Scott, 2003*). For instance, the mating conduct of male *Ae. aegypti* typically transpires indoors, in close proximity to the host, where both males and females are enticed. The application of irritant and repellent agents indoors, such as those inherent in insecticide-treated nets (ITNs) and indoor residual spraying (IRS), elicits responses from the vector characterized by attempts to escape the indoor environment due to the effects of irritation and repulsion. These responses may culminate in diminished indoor dwelling time, thereby diminishing the probability of male and female mating activities.

Previous studies have examined the conduct of male *Ae. aegypti* mosquitoes bearing a dominant lethal gene to insecticides. These investigations utilized a laboratory-based high-throughput screening technology (HITTS) to assess the mosquitoes' responses

(*Kongmee et al., 2014*). The current study investigated the impact of insecticides and repellents on the male SIT of *Ae. aegypti* during exposure, as this aspect has not been previously explored. This study aimed to assess the repellency response of the SIT strain to several substances, including citronella, DEET, deltamethrin, and transfluthrin. This was done by conducting a direct comparison of the male populations of *Ae. aegypti* with the USDA strain that escaped from an excito repellency (ER) chamber. The ER assay technique is a well-established behavioral testing system for evaluating sublethal chemical activities such as contact excitation and non-contact repellency of synthetic and naturally produced chemicals (*Sukkanon et al., 2022*). Insecticide-induced behavioral responses can be classified into two categories: irritation and repellency. Irritation occurs when an insect leaves a surface treated with the insecticide subsequent to tarsal contact with it. On the other hand, spatial repellency, also known as avoidance or deterrence, pertains to the ability of a compound to induce an avoidance response by compelling insects to move away from a chemical stimulus *via* direct physical contact, thereby diverting their attention away from the treated surface.

## MATERIALS AND METHODS

### Mosquitoes

*Aedes aegypti* (USDA strain) eggs were obtained from the United States Department of Agriculture, Gainesville, Florida, USA. The colony has been bred continuously under laboratory-controlled conditions at the Department of Entomology, Faculty of Agriculture, Kasetsart University, Thailand. This colony was physiologically susceptible to deltamethrin (*Juntarajumnong et al., 2012*) and transfluthrin (*Sukkanon et al., 2019*) using WHO adult bioassay (*WHO, 2016*). The larvae were reared in plastic pans ($20 \times 30 \times 8$ cm) containing 1,500 mL of tap water, with set population numbers per pan for synchronous development and were fed on standard commercial fish food pellets. Pupae were removed daily and placed in small cups until adult emergence in *screened* cages ($30 \times 30 \times 30$ cm) and were continuously supplied with 10% (w/v) sucrose solution *via* cotton sticks. Human blood was provided by using an artificial membrane feeding system (*Phasomkusolsil et al., 2017*). This protocol was approved (License No. U1-09598-2564) by the Kasetsart University Animal Ethics Committee and the Kasetsart University Institutional Animal Care and Use Committee, Bangkok, Thailand (Reference No. ACKU66-AGR-001).

### Pupae irradiation

Newly emerged *Ae. aegypti* male pupae up to age 1 day were irradiated according to *Kittayapong et al. (2018)*, at doses of 0 Gy (control, Non SIT) and 70 Gy (SIT) from colbalt-60 source of gamma ray ($\gamma$) emitting 1.47 KGy/h, in a Gamma Chamber 5000 (Board of Radiation and Isotope Technology, Department Atomic Energy, Mumbai, India) located at the Thailand Institute of Nuclear Technology (Public Organization), Nakhon Nayok. Then, plastic containers holding irradiated male pupae were placed in screened cages prior to adult emergence, with a 10% (w/v) sucrose solution provided *via* cotton sticks. The irradiated adult males aged 3–5 days were starved for 24 h before testing.

## Test compounds

**Plant essential oil (EO):** Citronella oil (*Cymbopogon nardus* (*L.*) *Rendle* essential oil; Lot. no: MK-40012 was extracted using steam distillation of citronella grass); was purchased from Thai- China Flavours and Fragrances Industry Co., Ltd., Ayutthaya, Thailand.

**Repellent:** DEET (N, N-diethyl-meta-toluamide; Lot. no: MKBH0428 V) with 97% active ingredient was obtained from Sigma-Aldrich®, Missouri, USA.

**Pyrethroid insecticides**

(1). Transfluthrin (2,3,5,6-tetrafluorobenzyl (1*R*,3*S*)-3-(2,2-dichlorovinyl)-2,2-dimethyl cyclopropanecarboxylate) with 97.90% active ingredient was provided by Sherwood Corporation (Thailand).

(2). Deltamethrin [(S)-alpha-cyano-3-phenoxybenzyl (1R,3R)-3-(2,2-dibromovinyl)-2, 2-dimethyl cyclopropanecarboxylate] with 98% active ingredient; Lot. no: DCM21512293) was provided by BASF Thailand.

## Filter paper treatment

Deltamethrin and transfluthrin were dissolved in mixture of acetone (Baker Analyzed™ A.C.S. reagent, J.T. Baker™, Fisher Scientific International, Inc., USA.) as an organic solvent and silicone as a carrier (Dow Corning® 556 silicon oil (Dow Chemical Company and Corning, Inc., MI, USA) to obtain doses of 0.0007% (*Juntarajumnong et al., 2012*) and 0.00852% (*Sukkanon et al., 2019*), respectively. Citronella and DEET were diluted to 2.5 and 5% in absolute ethanol (Merck, Darmstadt, Germany) based on a prior study demonstrating the optimal mosquito-repellent potential by *Nararak et al. (2016)* and *Sathantriphop et al. (2014)*. Subsequently, 2.8 mL of test solution was evenly spread over 14.7 cm × 17.5 cm sized filter papers (Whatman® No.1; Whatman International Ltd., Banbury, UK) using a 5 ml pipette controller following the method described by *Sathantriphop et al. (2014)*. Papers were prepared and used only once. Four similar sets of treated papers were prepared for each tested compound, whereas control papers were treated in the same manner using absolute ethanol or solvent mixture for insecticide tests only. All treated papers were air-dried in a horizontal position at room temperature for 1 h before the test (*Licciardi et al., 2006*).

## Contact irritancy (excitation) and non-contact repellency tests

The irritancy along with repellency responses of mosquito vectors have been assessed using ER chamber as described previously (*Bhoopong, Chareonviriyaphap & Sukkanon, 2022*; *Boonyuan et al., 2022*). This system consisted of two chambers with treatment-treated papers (one as a contact treatment chamber and the other for a non-contact treatment chamber) and two matched control chambers with control-treated papers (Fig. 1). In the contact chambers, four treated papers were placed in front of four inner screens, allowing mosquitoes to make direct physical contact with the treated areas. For the non-contact configuration, all four treated papers were placed behind the inner screens where mosquitoes could not make physical contact with the treated surface (to determine whether mosquitoes were repelled by smelling the airborne compound molecules inside the chamber). Fifteen irradiated male mosquitoes were introduced in each test chamber

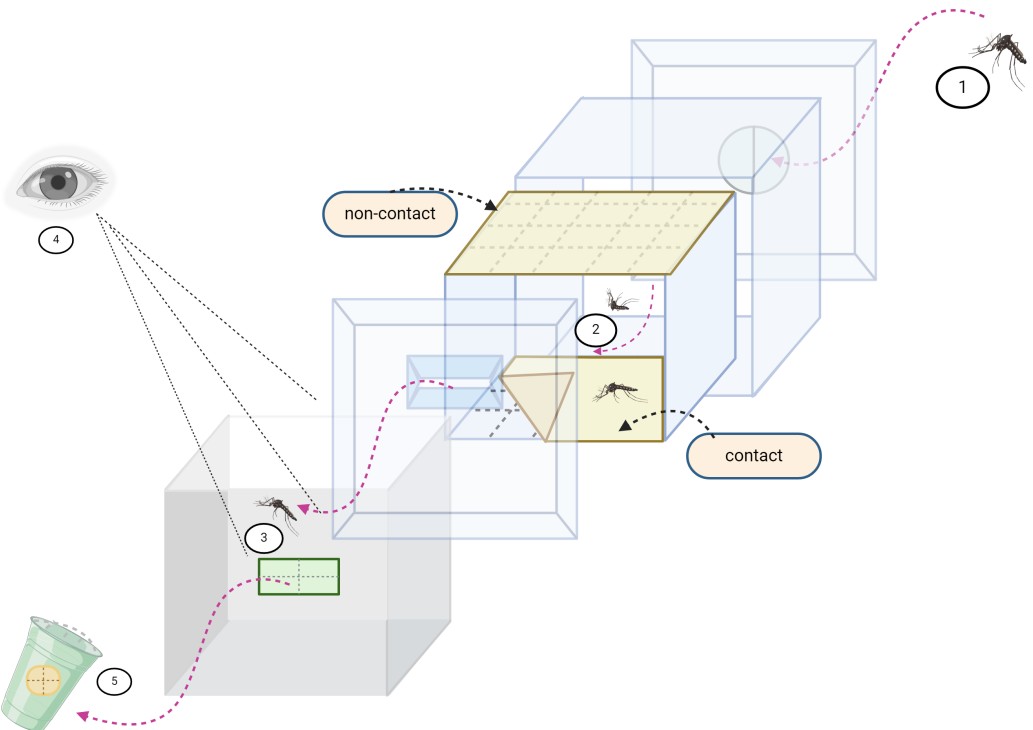

**Figure 1 Excito repellency schematic.** The excito-repellency assay system consists of the following components, summarized: (1) A total of 15 female mosquitoes were introduced through a rubber latex door; (2) The mosquitoes were given 3 min to acclimate inside the metal screen inner chamber. During this time, they were exposed to treated paper in two ways—with filter papers placed inside the inner chamber for a contact trial, and with filter papers positioned behind a mesh screen for a non-contact trial, aiming to regulate direct tarsal contact; (3) An exit portal was provided for mosquitoes to escape into the receiving passage box; (4) The experimenter recorded the number of escapes at 1-minute intervals using the naked eye; and (5) Mosquitoes that successfully escaped were collected into plastic holding cups.

and held for 3 min as an adjustment period to the chamber conditions. Then, the exit passage of each test chamber was opened to allow mosquitoes to escape from the test or control chamber to a receiving paper box connected to the chamber. Escaped mosquitoes were recorded and removed from the receiving box at 1 up to 30 min intervals (Fig. 1). After the exposure time, all remaining mosquitoes were removed from the chambers. The mosquitoes were kept in plastic cups and provided with 10% sucrose solution *via* cotton pads. Four replicates were required for each test compound. All tests were conducted during the period 0800–1600 h. Knockdown was recorded after the 30 min exposure time and for mortality at 24 h.

## Data analysis

Each trial was composed of four replicate paired treatment and control tests for contact and non-contact designs. The percentage of irradiated male mosquitoes that escaped from the test chamber was determined as follows: (total number of escaped mosquitoes/total number of tested mosquitoes) × 100. The mean percent escape and standard error of the mean (SEM) were then determined. The percentage of escape response were adjusted with

the paired controls to set the controls at zero using the Abbott's formular (*Abbott, 1925*) as follows: (% escape in test − % escape in control) / (100 -% escape in control) × 100. Statistical analysis of the ER assay was done according to *Roberts et al. (1997)*. Kaplan–Meier survival analysis was used to analyze and interpret the rate of escape of mosquitoes from each chamber at 1 min intervals (*Roberts et al., 1997*). The term "survivals" refers to the mosquitoes that remained in the test chamber from minute to minute, while those that managed to escape were classified as "deaths". The mosquitoes that were still in the exposure chambers at the conclusion of the test were labeled as "censored". A log-rank method was used to compare escape patterns of two mosquito condition for both the non-contact and contact trials (*Mantel & Haenszel, 1959*; *Bhoopong, Chareonviriyaphap & Sukkanon, 2022*). GraphPad Prism software (GraphPad Software, San Diego, CA, USA) was used for data analysis (*Sukkanon et al., 2022*). The mean escape percentage levels were compared statistically based on Duncan's multiple range test. A statistical significance for all tests was set at 5% ($P<0.05$).

## RESULTS

ER assay was used to evaluate the behavioral escape responses of SIT and Non-SIT strain of male *Ae. aegypti* exposed to citronella, DEET, deltamethrin, and transfluthrin for contact irritancy and non-contact repellency responses. The mean escape percentage from the contact and non-contact trials is presented in Fig. 2 and Table S1. In general, higher escape percentages were observed in the contact irritancy compared to non-contact repellency for all test compounds, except transfluthrin for the non-SIT strain. No significant differences ($P>0.05$) in the mean escape percentages were found between any of the control groups in either trial, regardless of test compound (Table S1). The results from the contact trials indicated that the responses of the SIT and Non-SIT strains to citronella at 2.5 and 5% (63.33–73.33% escape) were significantly greater than for DEET at 2.5 and 5% (26.67–31.67% escape) ($P<0.05$) and were not significantly different from deltamethrin (53.33–60.0% escape) ($P>0.05$). The lowest irritant effect was observed with DEET (26.67–31.67% escape) which was not significantly different from those caused by transfluthrin (28.33–45.0% escape). In the non-contact tests, citronella at 5% showed a repellency effect for SIT and Non-SIT strains (61.67% and 70% escape, respectively), with no significant difference to repellency by transfluthrin (40.0–48.33% escape), but significantly greater than DEET at 2.5 and 5% (8.33–13.33%). Furthermore, DEET exhibited the lowest performance in non-contact repellency; however, the percentage escape was not significantly different from those observed for deltamethrin (28.33–31.67), as shown in Fig. 2 and Table S1.

Knockdown and mortality of the escaped and non-escaped mosquitoes was only observed during the exposure period (30 min) with transfluthrin and deltamethrin, as shown in Fig. 3. The highest KD effect percentage (13.33–15% contact and 13.33% non-contact) was recorded from transfluthrin and deltamethrin in the SIT that failed to escape the exposure chambers in the contact and non-contact trials. In Non-SIT, the higher knockdown rate was recorded from non-escaped mosquitoes compared to mosquitoes that exited the treated chambers. The highest KD effect percentage (16.66% contact and 10.00–11.66%

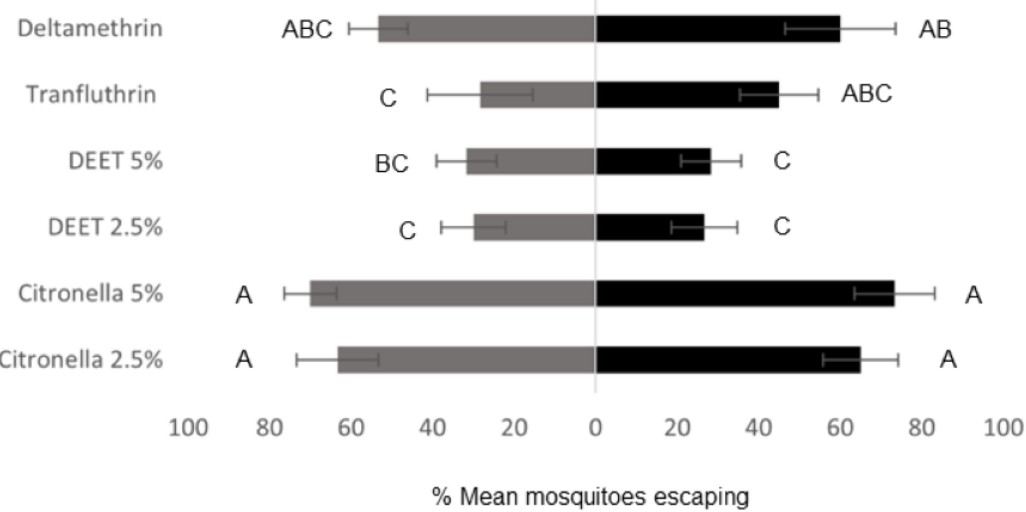

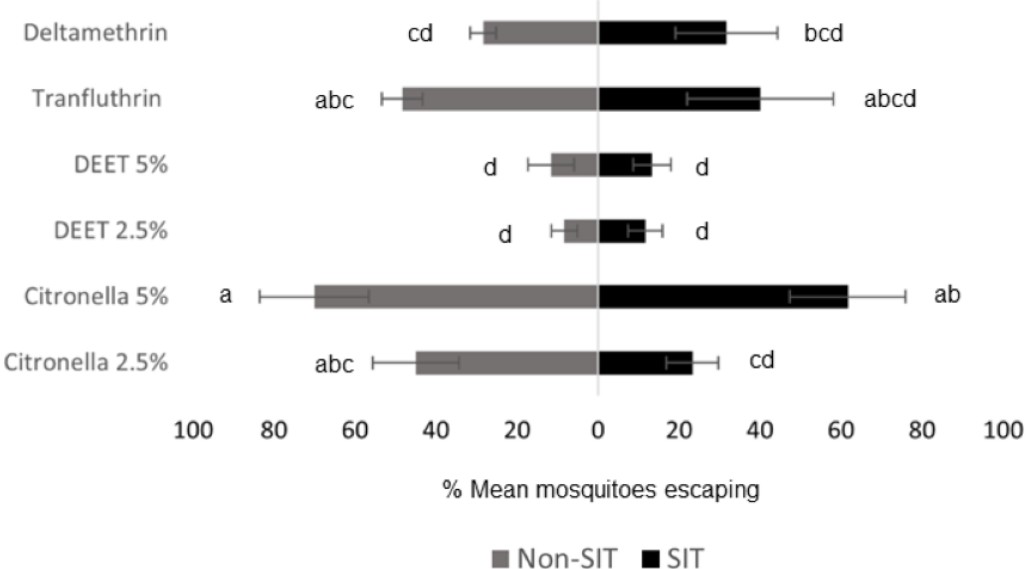

**Figure 2** **Mean percent escape response (% ± SE) of male *Ae. aegypti* (SIT and Non-SIT) at for citronella, DEET, deltamethrin, and transfluthrin at 30 min contact and non-contact exposures.** Letters above each bar indicate the Duncan's multiple range test groupings ($P < 0.05$). Bars with the same letter are not significantly different.

non-contact) was recorded from transfluthrin and deltamethrin. The levels of mortality for the male *Ae. aegypti* after the 24 h holding period were high for both the SIT and Non-SIT strains when exposed to transfluthrin; contact trials—SIT strain (26.66% for no escape), non-contact trials—SIT strain (20% for no escape), contact trials—Non-SIT strain (25%

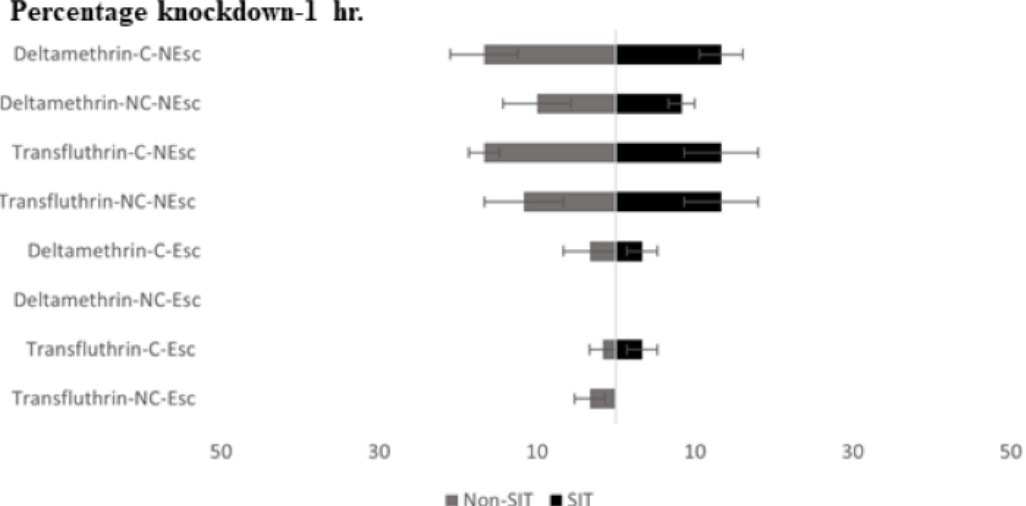

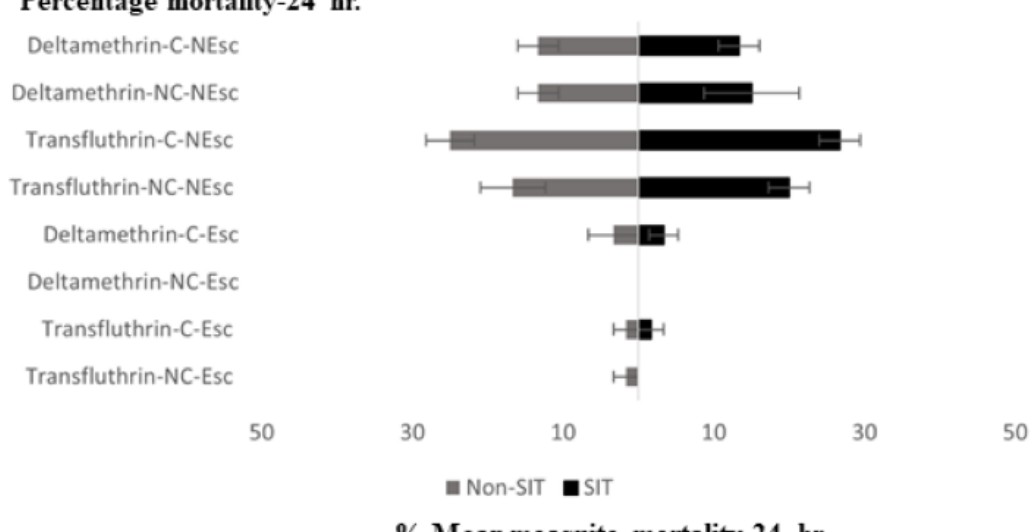

**Figure 3** Mean percent knockdown and mortality (% ± SE) of male *Ae. aegypti* (SIT and Non-SIT) at for citronella, DEET, deltamethrin, and transfluthrin at 30 min contact and non-contact exposures. C, contact; NC, non-contact; Esc, escaped mosquitoes; and NEsc, non-escaped mosquitoes.

for no escape) and non-contact trials–Non-SIT strain (16.66% for no escape). The control mortalities were low and did not exceed 5% in any test, as shown in supplementary data files.

The SIT and Non-SIT strains of the male *Ae. aegypti* exhibited different escape patterns in the contact and non-contact trials during the 30 min exposure period, depending on the test compound. Using survival statistics, probability patterns of the escape responses from contact and non-contact test chambers during 30 min exposure are shown in Figs. 4A–4F. These figures indicate the mean proportion of mosquitoes remaining in the ER chambers

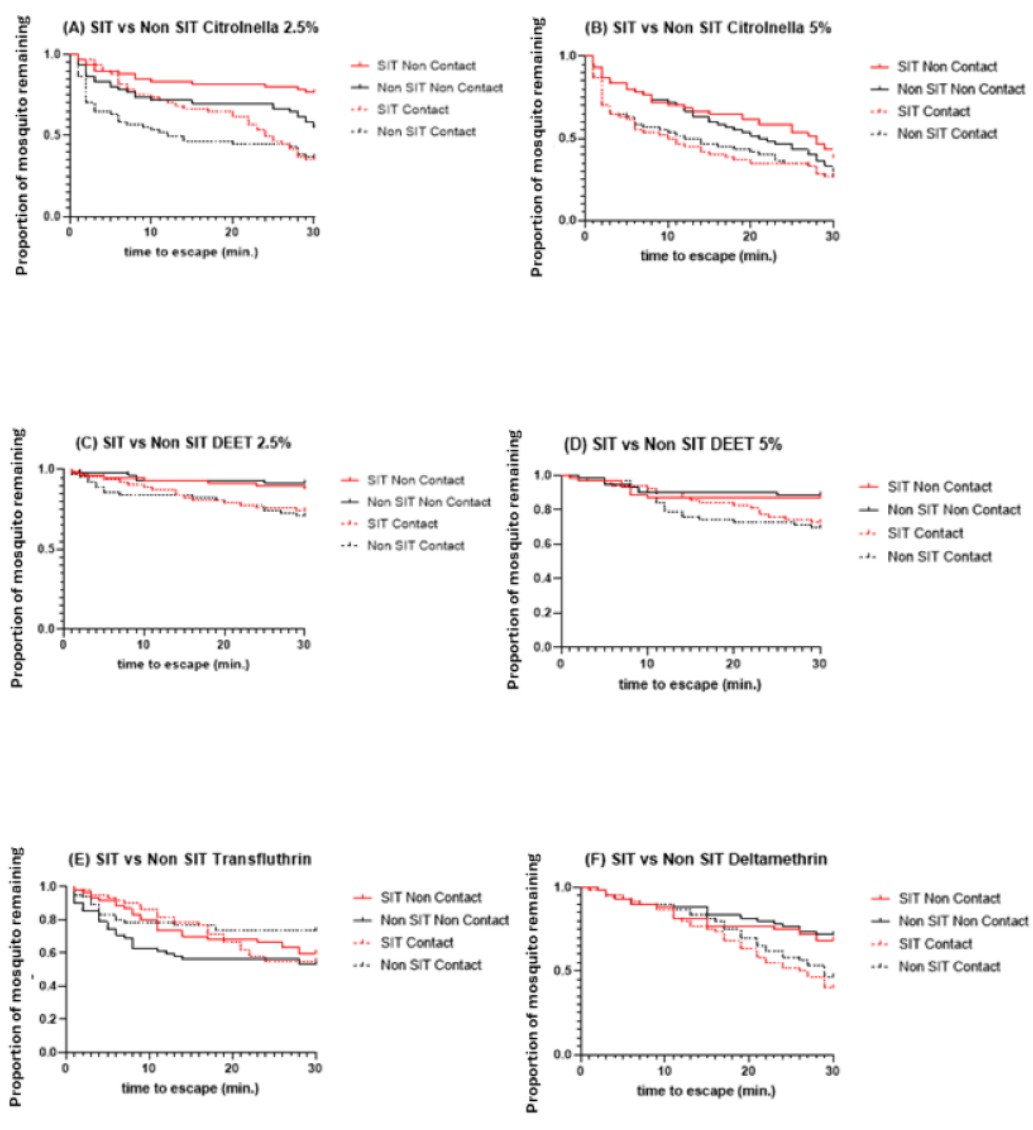

**Figure 4** **Proportion of mosquitoes remaining in excito-repellency chamber for radiation-sterilized male *Ae. aegypti* exposed to four test compounds (escape responses recorded at 1 min intervals during 30 min exposure period).** Paired control escape responses are not shown.

for 1 min intervals during the exposure time. The SIT strain had higher escape rates for 5% citronella (Fig. 4B) and deltamethrin (Fig. 4F) but these were not significantly different to the Non-SIT strain.

Statistical comparisons in the escape patterns between the SIT and Non-SIT strains exposed to citronella, DEET, deltamethrin, and transfluthrin in both contact or non-contact trials are shown in Table 1. There were no significant differences in escape patterns between both strains in both the contact and non-contact trials, except for the non-contact trial when exposed to citronella at 2.5%, where the levels were 23.33% and 45.0% escape in SIT and Non-SIT, respectively ($P = 0.0151$) were significantly different (Fig. 4A). When

**Table 1 Log-rank comparison of male *Ae. aegypti* escape patterns between SIT and Non-SIT strains exposed to citronella, DEET, deltamethrin, and transfluthrin in ER system.**

| Compound | Assay | |
| --- | --- | --- |
| | Contact | Non-contact |
| Citronella 2.5% | 0.4647 | 0.0151[*] |
| Citronella 5% | 0.8691 | 0.3196 |
| DEET 2.5% | 0.6761 | 0.5462 |
| DEET 5% | 0.6170 | 0.7681 |
| Transfluthrin | 0.2514 | 0.0923 |
| Deltamethrin | 0.4092 | 0.679 |

**Notes.**
[*]Indicates significantly different at $P < 0.05$ within test conditions using a chi-square statistic with 1 degree of freedom.

the contact and non-contact trials were compared for each strain, the SIT strain showed a significantly higher escape pattern in the contact trial than in the non-contact trial for DEET at 2.5% ($P = 0.0473$), for citronella at 2.5% ($P<0.0001$) and 5% ($P = 0.0351$), and for deltamethrin ($P = 0.0048$) and but there were no significant differences for DEET at 5% ($P>0.0771$) and transfluthrin ($P = 0.7118$). For the Non- SIT strain, there were significant differences among all compounds tested, except for citronella at 5% ($P = 0.2751$), as shown in Table 2.

# DISCUSSION

Currently, the most effective method for controlling vector-borne mosquito diseases is though chemical insecticides or repellents. However, a major challenge to the success of vector control programs is the emergence of insecticide resistance. As a result, scientists are under increasing pressure to develop improved tools for vector control (*Corbel et al., 2019*). To address this issue, alternative techniques are being explored, such as vector suppression, which involves the mass rearing and release of radiation-sterilized male mosquitoes to control or eliminate local populations (*Ritchie, 2014*). These techniques include the use of the SIT to control important disease-transmitting mosquito species to lessen the burden of mosquito-borne diseases on the global public health system (*Alphey et al., 2010*). Several research studies have integrated vector control approaches (*Achee et al., 2019*; *Golding et al., 2015*; *Van Den Berg et al., 2007*). Similar steps towards integrated vector management have been made in Thailand against dengue vectors (*Kittayapong et al., 2008*; *Kittayapong et al., 2019*; *Therawiwat et al., 2005*).

As mentioned above, Integrated Vector Management (IVM)—involves the utilization of a variety of interventions that can be scientifically demonstrated to be effective, either individually or in combination. The primary objective of IVM is to deploy control measures that are more economically efficient and to decrease dependence on any particular intervention strategy. Both chemical and nonchemical—have been used in a single endemic region; however, little is known about how these approaches interact when applied concurrently. For the purpose of comprehensive planning and disease prevention, it is essential to recognize any potential antagonistic (and synergistic) effects of one strategy

**Table 2** Log-rank comparison of male *Ae. aegypti* escape patterns between contact and non-contact chambers for SIT and non-SIT stains exposed to citronella, DEET, deltamethrin, and transfluthrin in ER system.

| Compound | Mosquito species | |
|---|---|---|
| | SIT | Non-SIT |
| Citronella 2.5% | <0.0001[*] | 0.0205[*] |
| Citronella 5% | 0.0351[*] | 0.2751 |
| DEET 2.5% | 0.0473[*] | 0.0035[*] |
| DEET 5% | 0.0771 | 0.0155[*] |
| Transfluthrin | 0.7118 | 0.0241[*] |
| Deltamethrin | 0.0048[*] | 0.0100[*] |

Notes.

[*]Significantly different at $P < 0.05$ within test conditions using a chi-square statistic with one degree of freedom.

on another. The comprehension of adult vector behavioral reactions to commonly used chemical control treatments is of utmost importance in order to formulate an effective deployment plan (*Wooding et al., 2020*). In the study of SIT technology, the success of the strategy depends on the irradiated male population's ability to mate with wild *Ae. aegypti* females. Inclusion of studies encompassing both sexes is crucial due to the potential influence of sex-related chemical disparities on disease dynamics and vector transmission (*Rutledge, Echano & Gupta, 1999*). For example, the chemical effects of repellents and irritants have the potential to interfere with or modify the house entry and indoor resting habits of *Ae. aegypti* mosquitoes. The results obtained from our investigations indicated that the SIT strain shows noteworthy contact irritancy in response to citronella, DEET, and deltamethrin. The implications of this study propose that in the event of successful evasion of insecticidal effects, SIT strains may engage in mating activities with female mosquitoes within their indigenous ecological setting. In addition, numerous observations have indicated that male *Aedes* mosquitoes are attracted to humans despite being incapable of feeding on blood. Observations in the field indicated that males swarm around and land on humans (*Amos, Ritchie & Cardé, 2020*; *Cator et al., 2011*; *Roiz et al., 2016*; *Visser et al., 2020*). Such evidence suggests that the mating process and subsequent transfer of genetic material could be negatively impacted.

The present study examined the behavioral responses of male *Ae. aegypti* strains (SIT and laboratory) to typical vector control agents in laboratory settings, focusing on irritating and repellent reactions. The findings of this study demonstrated that both the SIT and laboratory-reared males exhibited significant contact irritancy and non-contact repellency activity towards citronella, DEET, deltamethrin, and transfluthrin, when compared to the control group. These recorded behavioral responses align with previous research that investigated the behavioral responses of male populations of *Ae. aegypti* specifically developed for RIDL (Release of Insects carrying a Dominant Lethal) technology, as well as a study that examined a Malaysian wild-type population's reaction to chemical irritants and repellents using the HITTS method (*Kongmee et al., 2014*). The findings of the study revealed that there was a statistically significant ($P < 0.01$) manifestation of behavioral escape responses among all male *Ae. aegypti* test populations when subjected

to alphacypermethrin, DDT, and deltamethrin. This suggests that the anticipated irritancy and repellency behaviors were not inhibited by genetic modification. The most widely used active ingredients in conventional insect repellents are citronella oil and DEET (*Maia & Moore, 2011*). At concentrations range of 2.5–5% (v/v), DEET and citronella oil showed promising results on repellent activities against several female mosquitoes based on an ER test, with DEET, contact irritancy (26–30%) was significantly more effective than the 8–13% for non-contact spatial repellent assays (*Nararak et al., 2016*; *Sathantriphop et al., 2014*; *Tisgratog et al., 2016*). *Syed & Leal (2008)* studied sugar-feeding behavioral bioassay, showed that males also avoided DEET ($12.9 \pm 3.5\%$ of responding adults), with exposure to citronella eliciting both irritancy and repellency in the two test strains. This was consistent with the study by *Nakasen et al. (2021)*, which reported that *Cinnamomum verum* essential oil could repel both male and female *Culex quinquefasciatus* for up to 180 min. These consequences—repelling males as a side effect of a chemical's intended properties to deter female mosquitoes from striking—demonstrated that repellency is not sex-specific but rather a common adult behavior.

The magnitude of knockdown and mortality responses, represented by the percentage responses in the ER chamber, indicated that the SIT test cohorts demonstrated weaker responses compared to the female population. In this study, a decrease in escape was seen when male mosquitoes were permitted direct contact or non-contact with deltamethrin and transfluthrin, owing to the knockdown effect inside the chamber (non-escaping mosquitoes). No knockdown or mortality results were reported in sub-lethal ($LC_{50}$) doses of transfluthrin and deltamethrin against female *Ae. aegypti* (*Boonyuan et al., 2011*; *Sukkanon et al., 2020*). Similarly, *Allan (2011)* investigated five classes of insecticides (pyrethroids, phenylpyroles, pyrroles, neonicotinoids, and macrocyclic lactones) against *Cx. quinquefasciatus*, *Anopheles quadrimaculatus*, and *Ae. taeniorhynchus*. They reported that male *Cx. quinquefasciatus* were more susceptible than females. This difference may have been related to the larger size of the females than the male mosquitoes upon emergence (*Maciel-de Freitas, Codeço & Lourenço-de Oliveira, 2007*).

This study is limited by the exclusion of an assessment of the direct fitness parameters pertinent to SIT male *Ae. aegypti*, as delineated by *Bond et al. (2021)*. The findings elucidate a pronounced diminution in egg fertility associated with an augmented presence of sterile males within mating enclosures, resulting in an 88% decrease in the fecundity of both *Ae. aegypti* and *Ae. albopictus* in selected experiments. Consequently, a circumspect approach is imperative in the interpretation of the extant results. Subsequent investigations ought to center upon scrutinizing the repellency and irritancy effects of SIT mosquitoes in natural settings, concomitant with an exhaustive evaluation of additional direct fitness parameters.

In conclusion, a greater knowledge of the irritancy and repellency effects of chemicals is crucial for assessing the overall impact these compounds may have on both mosquitoes and disease transmission. These chemicals may interfere with normal behavioral host-seeking and blood-feeding activities. Based on the present data, it can be inferred that the male population of SIT exhibited a comparable response to regularly employed vector control agents, as observed in a laboratory setting, without any statistically significant variations. The current findings could be utilized to guide the implementation and analysis of early

SIT release trials in areas where additional treatments are being used. Further study should integrate fundamental studies of vector ecology (such as preferred mating times and locations) with controlled experimental evaluations of behavior in the field (such as hut studies using a mark-release-recapture design).

## ACKNOWLEDGEMENTS

We are grateful to Mr. Alex Ahebwa for his statistical assistance with this research. We would also like to thank Dr. Chutipong Sukkanon for the scientific feedback on this manuscript.

### Funding

The Thailand Institute of Nuclear Technology (Public Organization) and the Kasetsart University Research and Development Institute (KURDI), Bangkok, Thailand (Grant No. YF (KU) 33.65) provided financial support. This work was also supported by the Kasetsart University Research and Development Institute (Grant# FF (KU) 14.64). The funders had no role in study design, data collection and analysis, decision to publish, or preparation of the manuscript.

### Grant Disclosures

The following grant information was disclosed by the authors:
Thailand Institute of Nuclear Technology (Public Organization).
Kasetsart University Research and Development Institute (KURDI): YF (KU) 33.65.
Kasetsart University Research and Development Institute: FF (KU) 14.64.

### Competing Interests

The authors declare there are no competing interests.

### Author Contributions

- Wasana Boonyuan conceived and designed the experiments, performed the experiments, analyzed the data, prepared figures and/or tables, authored or reviewed drafts of the article, and approved the final draft.
- Amonrat Panthawong performed the experiments, analyzed the data, authored or reviewed drafts of the article, and approved the final draft.
- Thodsapon Thannarin performed the experiments, prepared figures and/or tables, and approved the final draft.
- Titima Kongratarporn performed the experiments, authored or reviewed drafts of the article, and approved the final draft.
- Vararas Khamvarn performed the experiments, prepared figures and/or tables, and approved the final draft.
- Theeraphap Chareonviriyaphap analyzed the data, prepared figures and/or tables, authored or reviewed drafts of the article, materials and equipment, and approved the final draft.

- Jirod Nararak conceived and designed the experiments, performed the experiments, analyzed the data, prepared figures and/or tables, authored or reviewed drafts of the article, and approved the final draft.

## Data Availability

The raw measurements are available in the Supplementary Files.

## Supplemental Information

Supplemental information for this article can be found online at http://dx.doi.org/10.7717/peerj.17038#supplemental-information.

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
