# Peer review of "Irritant and repellent behaviors of sterile male Aedes aegypti (L.) (Diptera: Culicidae) mosquitoes are crucial in the development of disease control strategies applying sterile insect technique"

_PeerJ, doi:10.7717/peerj.17038_

## Round 0.1 · original submission · Major Revisions

All three reviewers were positive about your work and its relevance. However, they also noted a number of important points that need your attention. In particular, a better explanation of the rationale behind the study as well as some methodological aspects. I hope that you will find their constructive comments useful in improving the revised version of the manuscript.

Reviewer 1 ·

Basic reporting

Pass

Experimental design

Fail
Experimental design is not clear (please see my full review).

Validity of the findings

Pass

Additional comments

Please see my full review (PDF file).

Annotated reviews are not available for download in order to protect the identity of reviewers who chose to remain anonymous.

Reviewer 2 ·

Basic reporting

The study of Boonyuan et al is well written, the language and grammar are also correct. The figures are clear and represent the data obtained. The tables are correct but can be improved.
The cited bibliography is adequate and up-to-date.
The raw data are adequate and well presented in the supplementary material.

Experimental design

My biggest concern is that it is difficult to understand, at least for me, the objective or rationale behind the work. Why was the evaluation carried out on males? If it is known that repellents are used to protect us from females that are the ones that feed on blood. If the SIT males are repelled to a lesser extent that NO-SIT by the substances evaluated in this work, this would favor the application of the SIT technique, since being less repelled would make it more likely that SIT males would mate with wild females. On the other hand, if the SIT males are more repelled by the evaluated substances than the wild ones, that would be a problem for the application of this technique. I think that the authors should focus more on these points to explain the importance of this work.
On the other hand, they use a laboratory strain to make the comparisons between SIT and NO-SIT. To make a real comparison, it should be done with a field population, since many times the behavior of a laboratory strain that has some inbreeding is very different from that of the field. A comment could be made about it.
Finally, I believe that supplementary table 2 complements the escape information in fig 1 because precisely in the treatments where there is knock-down or mortality by contact (transfluthrin and deltamethrin) mosquitoes do not escape. Is it possible that they don't escape because they are knock down? This possibility should be considered and discussed.

Validity of the findings

Some minor comments that require clarification
M&M. Pupae irradiation. How do they differentiate male from female pupae?
M&M. Filter paper treatment. Author state (lines 144-148)that: “Insecticides were dissolved in mixture of acetone (Baker Analyze A.C.S. reagent, J.T. Baker, Fisher Scientific International, Inc., USA.) as an organic solvent and silicone as a carrier (Dow Corning 556 silicon oil, Dow Chemical Company and Corning, Inc., MI, USA) to obtain doses of 0.0007% (Juntarajumnong et al. 2012) and 0.00852% (Sukkanon et al. 2019), 148 respectively.” What do you mean by respectively? Are they insecticides (transfluthrin and deltamethrin)? It is not clear if the concentrations of these insecticide are in a solution, but how much do they use to impregnate the paper? For repellents it is indicated that they use 2.8 ml, but for insecticides?
Why do the authors use a mixture as solvent (acetone: silicone) when using insecticides and only solvent (without silicone) when they use repellents (DEET, citronella)? This can change the bioavailability of the active ingredient (whether it is a repellent or an insecticide), it is known that they are generally hydrophobic and when the solvent evaporates, they will remain dissolved in the silicone oil while this would not happen with those papers that do not use silicone oil and therefore the bioavailability and entry through the cuticle may vary (especially for tests where there is contact with the active ingredient to be evaluated).
M&M. Contact irritancy (excitation) and noncontact repellency tests. I suggest adding a schematic so that the experimental design is better understood.
Table 1. It would be good to comment on how the escape pattern is calculated somewhere in the text since it is not clear, at least for me, the numbers shown in the table

Additional comments

There are some small grammar or typing corrections, for example (just to mention a few):

Abstract. Line 41, a comma is missing between DEET and transfluthrin.
Introduction. Line 62, “zika” should start with a capital letter
M&M. Lines 150-152 different color and font size.
Results. Fig 2 is very small; it is not easy to read.
All manuscript. Please verify that “aegypti” is written in lower case, many times when it appears after the point of Ae. it is automatically capitalized by autocorrect.
In the pdf Fig 1 and Fig 2 legends are incomplete or cannot be read. “Fig 1. Percentage escape response of male Ae. aegypti (SIT and Non-SIT) for citronella, DEET, transfluthrin and deltamethrin at 30 min contact and noncontact exposures. Using Duncan´s multiple range test, letters that differ indicate statistical di” (is not complete). “Figure 2. Proportion of mosquitoes remaining in excito-repellency chamber for radiation sterilized male Ae. aegypti exposed to four test compounds (escape responses recorded at 1 min intervals during 30 min exposure period). Paired control escape res” (is not complete).

·

Basic reporting

No comment

Experimental design

To test whether sterile male fitness is affected by radiation, Boonyuan et al. chose the escape response (ER) as a proxy measure of fitness and focused their study on the integrity of their behavioral ER to irritants and repellents. Overall, the ER assay used in this study showed no significant difference in the escape response between SIT and non-SIT male Ae. aegypti mosquitoes.
However, it is not clear why the authors chose this fitness parameter when competitive courtship and mating rates would have been a more direct and relevant measures. In fact, I would like to argue that it may be desirable that sterile males exhibit superior ERs, which would increase their competitiveness and survival in the wild. Although unsignificant, the authors observed such a difference to the benefit of the SIT strain with citronella and deltamethrin (lines 230-231). Significant repellent ERs were observed with statistical significance in favor of the SIT strain (see lines 238-240). Such resistant sterile strains would show increased success at entering houses (see lines 270-272) thereby outcompeting their fertile counterparts. I suggest these considerations are included in the discussion section.

Validity of the findings

No comment

Additional comments

Caption Figure 1. Correct spacing issues. End of caption is missing.

Caption Figure 2. End of caption is missing. Caption is lacking description for each panel (A through F).

Figure 2. Increase graph and legend size.

---

## Round 0.2 · accepted · Accept

Reviewers and I thank you for having improved the manuscript and responded to each comment. We recommend however paying attention and correcting typos and grammatical errors during the author proof-reading stage.

Reviewer 1 ·

Basic reporting

No comment

Experimental design

No comment

Validity of the findings

No comment

Additional comments

I think the manuscript was improved significantly. The authors addressed all my previous concerns and I don't have any new suggestions.

·

Basic reporting

The authors have addressed my concerns.
I recommend However that the manuscript is sent for copy-editing due to the presence of typos and grammatical errors.

Experimental design

No comments

Validity of the findings

No comments

Additional comments

None